# Behavioral and Cognitive Electrophysiological Differences in the Executive Functions of Taiwanese Basketball Players as a Function of Playing Position

**DOI:** 10.3390/brainsci10060387

**Published:** 2020-06-19

**Authors:** Yi-Kang Chiu, Chien-Yu Pan, Fu-Chen Chen, Yu-Ting Tseng, Chia-Liang Tsai

**Affiliations:** 1Institute of Physical Education, Health and Leisure Studies, National Cheng Kung University, Tainan 701, Taiwan; yigang3654@gmail.com; 2Department of Physical Education, National Kaohsiung Normal University, Kaoshiung 802, Taiwan; chpan@nknucc.nknu.edu.tw (C.-Y.P.); kiddlovejean@gmail.com (F.-C.C.); 3Department of Physical Education, National Tsing Hua University, Hsinchu 300, Taiwan; yuting75111@hotmail.com; 4Research Center for Education and Mind Sciences, National Tsing Hua University, Hsinchu 300, Taiwan

**Keywords:** cognition, inhibition, basketball, playing positions, Go/NoGo, event-related potential

## Abstract

The effect of the predominant playing position of elite basketball players on executive functions using both behavioral and electrophysiological measurements was investigated in the present study. Forty-six elite basketball players, including 27 guards and 19 forwards, were recruited. Event-related potential (ERP) signals were simultaneously recorded when the athletes performed the visual Go/NoGo task. Analyses of the results revealed that the guards and forwards groups exhibited comparable behavioral (i.e., reaction time (RTs) and accuracy rates (ARs)) performance. With regards to the electrophysiological indices, the guards relative to the forwards exhibited a shorter N2 latency in the Go condition, a longer N2 latency in the NoGo condition, and a smaller P3 amplitude across the two conditions. These results suggested that although the guards and forwards exhibited similar abilities in terms of behavioral inhibition, different neural processing efficiencies still exist in the basketball playing positions, with guards showing divergent efficiencies in the target evaluation and response selection of the target and non-target stimuli and fewer cognitive resources during premotor preparation and decision-making as compared to the forwards.

## 1. Introduction

Sports training can positively influence the brain related to neural functioning and cognitive performance [1,2,3]. Prolonged engagement within specific sports training processes and deliberate practice contribute to domain-specific expertise, resulting in elaborate cognitive performance [4]. However, the effects of sports training on cognition can vary across various types of sports due to the different amount of cognitive loads required for each sports type [3,5,6].

In general, a large range of sports can be roughly categorized into two types: open-skill and closed-skill sports [2,5,6,7]. Players in an open-skill sport are required to constantly adapt their actions and switch their strategies and skills according to the fast-changing, unpredictable environment [2,7]. These environments are usually externally paced [6]. On the contrary, closed-skill types of exercise take place in a relatively stable, predictable environment, in which performers behave at their own pace [5,6,7].

Considering the skills required in these two different domains, engaging in open-skilled sports requires various sets of motor skills (e.g., inhibiting movement from a proponent to an appropriate action in response to cues coming from teammates, opponents, and ball movements) and additional cognitive loads (e.g., planning and cognitive flexibility), while close-skilled athletes should require abilities related to attention maintenance in order to control their pace and concentrate on their own movements [6]. Therefore, open-skilled exercises participants experience greater cognitive loads and open-skilled sports affect cognitive abilities, especially executive functions, more than closed-skilled exercises [7]. Based on these concepts, different levels of executive functions can thus be modulated as a function of the different cognitive load requirements for various types of sports [6].

Executive functions (EF) are higher-order cognitive functions that regulate attention and actions, for which specific processes are especially necessary when individuals have to pay attention and concentrate in atypical situations [8,9,10]. Evidence has elucidated the effects of different sport types on inhibitory control via event-related potentials (ERPs) performance [5,11]. ERPs are defined as neural electrophysiological signals that reflect the brain’s electrical activity occurring during sensory, cognitive, or motoric processing [12,13]. The ERP N2 component is thought to reflect the conflict monitoring in early visual processing, which is a crucial process for early stage inhibitory control [14,15,16,17]. Di Russo et al. (2010) found that N2 latencies were significantly shorter for disabled basketball players compared to disabled swimmers [7]. However, basketball players did not significantly differ from healthy non-athletes. In addition, they also found that swimmers’ N2 amplitudes were significantly lower than those for basketball players and healthy non-athletes. The reduced N2 amplitudes and the slower N2 latencies shown by swimmers were interpreted as difficulty related to inhibiting intended actions. By contrast, the fact that basketball players’ N2 amplitudes and latencies corresponded with healthy non-athletes suggested that open skill sports training increases inhibitory ability by compensating for a disability [7]. Given the superior inhibitory controls of open-skilled athletes, the finding provided evidence that open-skilled training can benefit executive functioning [5,7].

However, playing positions in open-skilled sports features specific tasks, roles, and jobs [18]. For players to exhibit successful performance in a specific position, extended exposure to practices that conform to the demands of specific tasks are necessary. These procedures ultimately contribute to expert performance (i.e., physical skills) associated with a specific playing position [4]—that is, position-specific expertise [19]. In addition, the development of position-specific expertise is due to the neurocognitive adaptation of the cognitive function that characterizes such expert performance [4]. Therefore, it is the cognitive differences that can be differentiated among playing positions, which is in keeping with the differences in positional behavioral patterns.

Studies have confirmed that diverse cognitive styles exist between playing positions in open-skilled sports. For example, Montuori et al. (2019) investigated positional effects on cognitive flexibility in elite volleyball players and reported that mixed players had a longer reaction time (RT) and higher accuracy rates (ARs) compared to strikers and defenders [10]. Another study conducted by Schumacher et al. (2018) investigating the potential effects of general attention, anticipation, and reaction in soccer players with different playing positions found that midfielders only showed a significantly faster reaction time (RT) in an acoustic RT test (ART) as compared to defenders [20]. These findings supported the supposition that the executive profile in open-skill sports is position-specific. In addition, Vestberg et al. (2017) suggested that, in soccer, attackers require stable executive functions, while midfielders require impulsivity and defenders need inhibitory control [21]. Despite the evidence demonstrating that playing positions induce divergent cognitive benefits, these positional effects may vary across different types of open-skilled sports due to the nuances specific to a type of sport [3,11]. Accordingly, more studies that investigate the positional effects of open-skilled sports other than volleyball and soccer are still needed. Most importantly, previous works regarding this issue were primarily focused on behavioral performance, in which instantaneous shifts in attention and the consequences of visual stimulation processes cannot be observed independently [1,5]. Thus far, the neural mechanism that modulates the executive functions as a function of playing position is still unclear. Hence, the further exploration of positional effects on cognition in open-skilled athletes is necessary.

Basketball is a team sport that consists of two teams opposing one another. Teams compete for primary goals by making successful shots with a ball through their opponents’ hoop and preventing their opponents scoring through their own. Hence, physical skills or strategies related to scoring and defense, tactical execution, and team cooperation are crucial for basketball players [19]. During basketball matches, players are continuously bombarded with both relevant and distractive information that may potentially imperil their response execution [22]. These contrasting sources of information put substantial loads on the executive abilities of basketball players that require critical focus, carrying out actions, and concurrently controlling attention while being distracted by conflicting information (i.e., inhibitory controls) [22].

In general basketball competitions, basketball players can be precisely categorized into five positions: (1) point guard (PG), (2) shooting guard (SG), (3) small forward (SF), (4) power forward (PF), and (5) center (C). Traditional classifications tend to further classify players into guards (SG and PG), forwards (SF), and centers (PF and C) due to the similarities in their assignments in a match [18,19]. Since physical skills can differentiate playing positions in basketball [19,23], we hypothesized that there would be executive differences in relation to behavioral inhibition exhibited between basketball player positions due to the contrasting physical skill sets that underlie inhibitory controls. For example, penetration requires instant responses to external cues [24], where penetrators have to suppress their ongoing movement (e.g., moving direction, skills, or velocities) in response to defenders and execute more effective movements. In the analysis of the physical skills required for successful performance in a given playing position, penetration movements are executed by players in all positions, while guards penetrate primarily facing the basket and centers penetrate with their back to the basket [19]. It is worth noting that contrasting differences exist between facing or backing penetration. Penetration facing the basket requires instant execution and the suppression of movements, emphasizing both velocity and immediate responses, while penetration with the back to the basket requires powerful force to squeeze into the painted area, stressing physical collision and lower limb stability [18]. Additionally, centers execute more static tasks (e.g., gaining position, blocking, and screening) than guards, for which the movements require a significant amount of physical resistance and contact [19], with relatively fewer uncertain factors as compared to penetration. In addition, the evidence of physical activities can also provide the foundations for positional differences. Basketball exercises generally include substantial changes of direction in response to environmental stimuli; these changes may be related to information processing that involves visual scanning, decision-making, and reactive movements [25]. However, substantial environmental-stimuli from frequent directional changes may positively affect the cognitive abilities of basketball players. Previous studies have reported that guards tend to execute more frequent movement and directional changes than centers [24,26]. Given the greater numbers of changes in directions, accelerations, and decelerations executed by guards when performing at a higher movement speed than centers, the cognitive loads suffered by guards may be therefore greater than those imposed on centers. Taken together, differences in physical skills may potentially modulate the inhibitory control of basketball players based on the different playing positions.

Inhibitory control requires the control of the attentional ability to be unaffected by a strong internal inclination or an external temptation, making it possible to control actions; suppress automatic, dominant, or prepotent responses; and instead act appropriately [8,9,14]. Inhibitory ability is rather critical to athletes due to a positive association with successful performance in open-skilled experts, so successful players can make quick decisions while instantly inhibiting planned movements [8,9,22]. Such a cognitive function is necessary to the successful performances of basketball players in practical competitions and can be differentiated based on basketball playing positions. Accordingly, determining whether inhibitory controls may be modulated by different playing positions owing to several position-specific features of physical skills and motor executions needs to be explored.

The Go/NoGo task is a perceptual discriminative task that is frequently used to investigate inhibitory ability [14,27], especially in elite athletes [7,11,28]. In this paradigm, an individual has to produce a speedy covert response to a target stimulus and must refrain from responding to a non-target stimulus [14,27,29,30]. The procedures in the task mimic the complex visual motor behavior patterns of basketball players, including task preparation, stimulus identification and evaluation, response selection, and response execution or inhibition [7,30].

As mentioned above, ERP signals have been used to detect the dynamic brain activity reflecting instant neural cognitive processing [12,13]. Its excellent temporal resolution allows for determining the time sequences of neural and psychological processes from millisecond to millisecond [12], which behavioral measurement techniques leave blank [27,31]. ERP recordings reflect the timing and organization of several information processes in the brain’s networks. These sensitive indices have been widely applied to evaluate cognitive differences across exercise groups [3,6]. Two ERP components (e.g., N2 and P3) are activated when performing the Go/NoGo task coupled with ERP recording. The N2 components are defined as the peak negative voltage evoked after the appearance of the stimuli during a time window of 200–350 ms, whereas the P3 components are the positive deflection that peaks after the appearance of the target in a time window of 350–600 ms [15]. In addition, N2 and P3 can reflect the mental processes involved in conflict monitoring [14,15,16,17], as well as premotor evaluations and decision-making [2,3,15,27,29,30,32], respectively.

Notwithstanding the fact that previous studies have revealed divergent executive profiles across playing positions in open-skill sports [10,20], extended investigations are still needed for the following reasons: (1) nuances that are different across various types of open-skilled sports may cause diverse positional effects on cognition [3,11]; and (2) previous findings only emphasized behavioral performance [10,20], leaving the actual neuronal mechanism still unclear. To the best of our knowledge, no research has yet been conducted to demonstrate inhibitory ability as a function of basketball playing position using a Go/NoGo paradigm coupled with neural electrophysiological techniques. Therefore, the aim of the present study was to investigate the effects of predominant playing positions on inhibitory controls using both behavioral (i.e., RT and AR) and electrophysiological (i.e., N2 and P3 latencies and amplitudes) measurements in elite basketball players when performing the visual Go/NoGo task. In the present study, basketball players were classified into two groups: guards (i.e., PG, SG, and SF) and centers (i.e., PF and C), given that this classification is the one that experts and coaches most commonly adopt [18,19]. We hypothesized that significant positional effects would be observed, showing that guards exhibit superior performance in neurocognitive (i.e., behavioral and electrophysiological) measurements as compared to forwards.

## 2. Materials and Methods

### 2.1. Participants

Forty-six male elite players (mean age 20.57 ± 1.13 years) participating in the highest level of basketball competitions in Taiwan were recruited from six University Basketball Association (UBA) basketball teams in this study. Since previous research has revealed that differences in neuropsychological and neurophysiological characteristics exist between genders [33] and that differences in training loads also occur between male and female basketball players [34], the experimental design of mixed-gender groups may result in unbalanced neuropsychological changes regarding the cognitive measurements [33]. Only male players were thus recruited in the current study. Twenty-seven guards and nineteen forwards were included. The playing positions of the participants were self-reported and reported by the coaches, and those who played multiple positions (e.g., across both guards and forwards, especially between SF and PF) were excluded from this study. Both of the groups were matched for age, level of education, and hand dominance. All the players were non-smokers, with normal or corrected-to-normal vision. In addition, they were free of cardiovascular, psychiatric, metabolic, or neurological diseases, as well as any history of head injuries. Participants who were taking medications that could influence the function of the central nervous system were excluded. The participants were instructed to avoid caffeine, alcohol consumption, and moderate-to-vigorous workouts on the day before the experiment. None of the players exhibited cognitive impairment, as measured by scoring above 26 on the Mini-Mental-State Examination (MMSE) [35]. Written informed consent, which was approved by the Institutional Ethics Committee of National Cheng Kung University (REC 108-567), was obtained from all of the players after a full explanation of the experimental procedure. All of the players obtained payment for their participation.

### 2.2. Procedure

The participants were asked to visit the cognitive neurophysiology laboratory twice. On the first visit, a demographic questionnaire; an informed consent form; and the MMSE and Beck Depression Inventory, 2nd edition (BDI-II) [36], assessments were completed, and BMI was calculated (weights/heights^2^) to avoid confounding effects on cognition [37,38]. The cognitive task was performed in an acoustical shield laboratory with dimmed light. The participants were seated in front of a color computer monitor (with 47.5 × 27 cm) placed approximately 70 cm from the eyes, and an electrocap and electro-oculographic (EOG) electrodes were attached to their faces and scalps before the Go/NoGo task initiated. On the second visit in the same week, the values of the estimated VO_2_ max were measured using a Yo-Yo intermittent recovery test (Yo-Yo IR1) [39,40].

### 2.3. Go/NoGo Paradigm

In the Go/NoGo paradigm, either a blue or yellow dot was presented as the visual stimuli using the E-prime 2.0 neural stimulation system (Psychology Software Tools, Inc., 311 23rd Street Ext. Suite 200 Sharpsburg, PA, USA). The stimulus was presented in the center of the computer monitor at a visual angle of 2.5 degrees. There were four blocks with 60 Go (blue dots) and 20 NoGo (yellow dots) stimuli in each trial. The participants were required to respond to the Go stimuli as quickly as possible by pressing the space button with their dominant hand and to refrain from responding when the NoGo stimuli appeared. The order of presentation was random within blocks and across participants. The entire task procedure is illustrated in Figure 1. Each trial was initiated, and a white cross served as the fixation point. The first trial was initiated with the instructions, followed by a 3 s countdown and a 1000 ms period prior to the commencement of the fixation cross. Following the offset of the fixation, either a Go or NoGo stimulus was presented. The Go stimulus was presented for 500 ms, while the NoGo stimulus lasted for 3000 ms. The next trial was initiated once the space button was pressed (in both the Go and NoGo conditions) or withheld for 3000 ms (in the NoGo condition). Consequently, various intertrial intervals were randomly presented at between 1100 and 1700 ms. Before recording the electrophysiological signals, the participants performed practice blocks to ensure that they were familiar with the task. During the task, the participants had to constantly focus on the center of the monitor to avoid body movements and saccades. A duration of two minutes was given to every participant to rest between blocks.

### 2.4. Cardiorespiratory Fitness Estimation

The Yo-Yo IR1 test was conducted to estimate cardiorespiratory fitness (i.e., VO_2max_) according to procedures described previously (Krustrup et al., 2003). The test consists of repeated shuttle runs (2 × 20 m) that are performed at a progressively increasing velocity interspersed with active recoveries (2 × 5 m, 10-s) until exhaustion. The test was terminated when the players failed to complete the distance on time twice, or they subjectively felt that they were unable to maintain the speed. The total distances covered during the test was considered the players’ scores [39]. After measuring the final score of the test, the estimation of VO_2_ max was quantified using the following formula: VO_2_ max (ml kg^−1^ min^−1^) = players’ scores (m) × 0.0084 + 36.4 [40].

### 2.5. Electrophysiological Recording and Analysis

Electrophysiological data were recorded from participants when they were performing the Go/NoGo task using the Syn-Amps Electroencephalography (EEG) amplifier and the Scan 4.5 package (Neuroscan Inc., El Paso, TX, USA) with 23 electrodes (including Fz, F3, F4, FCz, FC3, FC4, Cz, C3, C4, CPz, CP3, CP4, Pz, P3, P4, PO3, PO4, PO7, PO8, Oz, O1, and O2) according to the International 10-20 System. All the electrodes were referenced to linked bilateral mastoid electrodes. An electrode placed at the mid-frontal scalp served as the ground. The horizontal and vertical electrooculographic (EOG) activities were bipolarly recorded with two pairs of electrodes placed 2 cm from the canthi of both eyes and 2 cm above and below the left eye, respectively. All the electrode impedances were maintained below 5Ω. All the EEG data were digitized with an analog-to-digital (A/D) rate of 500 Hz/channel; amplified with a band-pass filter of 0.1–30 Hz (Zero Phase Shift filtering mode will be adopted), including a notch filter of 60 Hz; and stored for an off-line analysis using the Scan 4.5 analysis software (Neuroscan Inc., El Paso, TX, USA).

Trials with missed Go responses, false NoGo responses, and RTs faster than 150 ms and slower than two standard deviations were excluded from the analysis. The EEG data were segmented into epochs starting from −200 ms prior to and 600 ms after the stimulus onset. Consequently, the epoch trials that were contaminated by ocular artifacts with amplitudes exceeding ± 100 mV were discarded from the analysis. Clean epoch trials were then baseline corrected, averaged to form a grand ERP, and constructed according to various conditions.

Two major stimulus-locked ERP components, N2 and P3, were measured at the Fz, FCz, and Cz electrodes. The mean N2 amplitude and latency were measured within a time window of 200–350 ms, and the mean P3 amplitude and latency were measured within a time window of 350–600 ms [15].

### 2.6. Statistical Analysis

Between-group comparisons (guards vs. forwards) of the demographic data including age, level of education, height, weight, BMI, estimated VO_2_ max, scores on the MMSE and BDI-II, years of playing basketball experience, training frequency per week, and training period were analyzed using an independent t test. Hand dominance was subjected to a Pearson’s chi-squared test. A demographic index (e.g., MMSE, BDI-II, BMI, and VO_2max_) served as a confounding variable whenever a significant difference occurred between the two groups [37,38]. RTs were subjected to an independent t test, and the ARs were subjected to a repeated-measured 2 (group: guards and forwards) × 2 (condition: Go and NoGo) ANOVA, with the group as the between-subjects factor and the condition as the within-subjects factor. The electrophysiological indices were subjected to a repeated-measured 2 (group: guards and forwards) × 2 (condition: Go and NoGo) × 3 (electrode: Fz, FCz, and Cz) ANOVA, with group as the between-subjects factor and condition and electrode as the within-subjects factors. A Bonferroni post hoc analysis was performed when a significant difference occurred. A Greenhouse–Geisser correction was adopted to adjust the significance level when the sphericity assumption was violated. The level of significance was set at *p* < 0.05.

## 3. Results

### 3.1. Demographic Characteristics

Table 1 provides an overview of the demographic characteristics of the guards and the forwards. The two groups were matched at the group level in terms of age, handedness, BMI, and VO_2_ max. Basketball experiences, training frequency, and training period were also comparable between the two groups. There were no significant differences in the level of education, mental state, and depression. There were, however, significant between-group differences in height and weight, with forwards being taller (*t*_(1)_ = −7.495, *p* < 0.001) and heavier (*t*_(1)_ = −4.561, *p* < 0.001) than guards.

### 3.2. Behavioral Performance

As illustrated in Figure 2, no significant between-group differences in RT (*t*_(44)_ = −0.42, *p* = 0.679) were observed. A significant main effect of condition on the ARs (*F*_(1,44)_ = 8.77, *p* = 0.005, *η**_p_*^2^ = 0.17) was observed, indicating higher ARs in Go conditions (98.08% ± 3.06%) compared to in NoGo conditions (95.94% ± 3.36%) across the two groups. No significant main effects of group or condition x group interactions were observed.

### 3.3. Electrophysiological Performance

In Figure 3, the grand-average ERP waveforms obtained from the three midline electrodes (e.g., Fz, FCz, and Cz) in the two groups are reillustrated.

#### 3.3.1. N2 Latency

There were significant main effects of condition (*F*_(1,44)_ = 13.51, *p* = 0.001, *η**_p_*^2^ = 0.24) and electrode (*F*_(1.75,76.982)_ = 18.81, *p* < 0.001, *η**_p_*^2^ = 0.47) on the N2 latencies, indicating that the N2 latencies in the NoGo condition (271.71 ± 2.49 ms) were shorter than those in the Go condition (283.67 ± 3.50 ms) across the two groups. Electrodes on the N2 latency were observed to have the following gradient: Cz (274.67 ± 22.69 ms) < FCz (277.20 ± 21.30 ms) < Fz (280.85 ± 21.47 ms). There was also a significant effect of the group x condition interactions (F_(1,44)_ = 7.58, *p* = 0.009, *η**_p_*^2^ = 0.15). Post hoc comparisons showed that, in comparison to the forwards, the guards had shorter N2 latencies in the Go condition (guards vs. forwards: 277.64 ± 24.72 vs. 288.82 ± 22.76 ms, *p* = 0.007) and longer N2 latencies in the NoGo condition (guards vs. forwards: 274.54 ± 18.33 vs. 267.89 ± 16.49 ms, *p* = 0.030).

#### 3.3.2. N2 Amplitude

There were significant main effects of electrode (*F*_(1.466,22.273)_ = 22.27, *p* < 0.001, *η**_p_*^2^ = 0.34) and condition (*F*_(1,44)_ = 107.68, *p* < 0.001, *η**_p_*^2^ = 0.71) on the N2 amplitudes, indicating that the N2 amplitudes in the NoGo condition (−6.90 ± 6.17 μV) were larger than those in the Go condition (0.44 ± 4.12 μV) across the two group and three electrodes. Electrodes on the N2 amplitudes were observed to have the following gradient: Fz (−4.33 ± 5.36 μV) > FCz (−3.66 ± 6.55 μV) > Cz (−1.70 ± 6.95 μV). There were also significant effects of condition × electrode interactions (*F*_(1.752,77.096)_ = 3.84, *p* = 0.031, *η**_p_*^2^ = 0.08). A post hoc comparison showed that N2 amplitudes in the Fz (−1.08 ± 3.32 μV) were significantly larger than those in the Cz (2.12 ± 4.54 μV) in the Go condition.

#### 3.3.3. P3 Latency

A significant main effect of electrode on the P3 latency (*F*_(1.624,71.442)_ = 11.01, *p* < 0.001, *η**_p_*^2^ = 0.20) was observed, with the following gradient: Fz (381.42 ± 26.17 ms) > FCz (377.97 ± 28.85 ms) > Cz (377.10 ± 27.24 ms). There were also significant effects of group × electrodes interaction on the P3 latencies (*F*_(2,43)_ = 6.15, *p* = 0.003, *η**_p_*^2^ = 0.12). A post hoc comparison showed that the P3 latencies in the Fz (380.87 ± 23.16 ms) were slower than those in the FCz (373.82 ± 24.58 ms) and Cz (372.24 ± 23.99 ms) in the forwards group.

#### 3.3.4. P3 Amplitude

There were significant main effects of condition (*F*_(1,44)_ = 282.29, *p* < 0.001, *η**_p_*^2^ = 0.87) and electrode (*F*_(1.254,55.183)_ = 50.42, *p* < 0.001, *η* = 0.53), indicating that the P3 amplitudes in the NoGo condition (17.32 ± 6.8 μV) were larger than those in the Go condition (7.8 ± 5.36 μV) across the two group and three electrodes. The electrodes for the N2 amplitudes were observed to have the following gradient: Cz (14.14 ± 7.78 μV) > FCz (13.22 ± 8.1 μV) > Fz (10.31 ± 6.88 μV). A significant main effect of group (*F*_(1,44)_ = 5.34, *p* = 0.026, *η**_p_*^2^ = 0.11) was also observed, with the P3 amplitudes being larger in the forwards group (14.64 ± 8.56 μV) than in the guards group (11.10 ± 6.78 μV) across both conditions and electrodes. A significant effect of the conditions x electrodes interaction (*F*_(1.637,72.02)_ = 10.07, *p* < 0.001, *η* = 0.19) was observed. A post hoc comparison showed that the P3 amplitudes at Cz (9.50 ± 5.6 μV) were significantly larger than those at Fz (5.82 ± 4.33 μV) in the Go condition, and the P3 amplitudes at FCz (18.74 ± 6.99 μV) and Cz (18.78 ± 6.86 μV) were significantly larger than those at Fz (14.81 ± 5.91 μV) in the NoGo condition.

## 4. Discussion

In the present study, we investigated the effects of predominant playing positions on inhibitory controls using both behavioral (i.e., RT and AR) and electrophysiological (i.e., N2 and P3 latencies and amplitudes) measurements in basketball players when the participants were performing the visual Go/NoGo task. The main findings of the present study were as follows: (1) a comparable behavioral performance (i.e., RTs and ARs) was found between the guards and forwards groups, (2) the guards relative to the forwards had a shorter N2 latency in the Go condition and a longer N2 latency in the NoGo condition, (3) the guards group exhibited a smaller P3 amplitudes across the two conditions and three electrodes than the forwards group.

### 4.1. Behavioral Indices

The present findings, indicating that the guards and forwards groups had comparable RTs and Ars, were inconsistent with the findings of previous works [10,20]. Montuori et al. (2019) suggested that mixed players in volleyball had longer RTs and higher ARs than strikers and defenders when engaged in a sport-specific task-switching task [10]. Schumacher et al. (2018) revealed significant behavioral differences in a visual RT (VRT) task between soccer midfielders and strikers [20]. The lack of differences in behavioral performance was unexpected. We predicted that guards might exhibit a superior behavioral performance compared to the forwards due to the cognitive adaptation resulting from higher levels of cognitive load resulting from their position-specific expertise [19,23] and additional changes in direction while playing basketball [26]. However, this prediction was not consistent with our current findings. Possible interpretations of the comparable behavioral performance may thus be proposed.

Firstly, the way to distinguish the playing positions may be different between basketball and volleyball [10]. In volleyball, players seldom execute tasks that are beyond their positional domain. In other words, attackers seldom play defense, and, conversely, defenders seldom play attackers [10]. Defensive players engage in a judgement that requires continuous reactions to the misdirection and distractions created by the offensive teams [4]. They have to continuously read the intentions of their opponents, process their deceptive actions, and prevent their offensive maneuvers [4,22]. On the other hand, offensive players behave in a selective rather than a passive way due to their possession of the ball. Since they can decide what movements they want to take, offensive players usually play with less uncertainty when executing their decisions because of a lack of illusive schemes from the defensive side [4]. Taken together, given that the tasks required for the playing positions are primarily based on offensive and defensive characteristics, we can tentatively conclude that positional differences in behavioral performance in volleyball are due to the contrasting expertise that is necessitated in offensive vs. defensive schemes. These concepts were supported by investigations of cognitive differences between other offensive and defensive open-skilled players. Wylie et al. (2018) found that defensive football players assume lower interference costs when conducting the Erickson flanker task in comparison to offensive players [22], and Williams et al. (2008) found that defensive soccer players display greater anticipation skills than offensive players in both defensive and offensive situations [4]. Nevertheless, unlike the open-skilled sports mentioned previously, differentiation among basketball positions is based on anthropometrics. In other words, taller, heavier players (as also seen in the present study) are usually placed in the painted area (i.e., forwards), and shorter, lighter players are located in the perimeter area (i.e., guards) [18,19]. It is necessary to mention that task demands based on offensive or defensive plans cannot directly help distinguish between basketball playing positions. Conversely, basketball players all participate in offensive and defensive actions when engaged in the sport. On the offensive side, guards assist, penetrate [23], and perform long-distance shooting [19], and forwards set screens, attack with their back to the basket, and play supportive roles [19]. On the defensive side, guards play one-on-one defenses and prevent their opponents from penetrating their defense [19], and forwards are primarily responsible for blocking, rebounding [23], and cooperative defending [19]. Taken this into consideration, although the division of playing positions on the basis of offensive and defensive schemes served as the primary foundation for interpretation in previous studies, it seems that the effect of different schemes cannot be applied to basketball because of the unique playing position differentiation. As a result, the lack of observation of behavioral differences may be in relation to universal participation in both offensive and defensive tactics for basketball players in all positions, and to a lesser extents, guards and forwards.

The other possible explanation for the inconsistency might be that the positional effect in open-skilled sports on cognitive functions appeared to be selective. In a previous work, Schumacher et al. (2018) found that different playing positions among soccer players could only distinguish RT behavioral performance and could not be used as an indices for sustained attention and anticipation, implying that a selective cognitive advantage can be observed in open-skilled sports [20]. In the present study, only the visual Go/NoGo task was adopted to assess the behavioral differences in executive functions in guards and forwards. Further research is warranted in this area, possibly examining the positional effects on different cognitive domains in basketball.

### 4.2. Electrophysiological Performance

The present results showed the electrophysiological differences in N2 latency in both the Go and NoGo conditions and the P3 amplitude across the Go and NoGo conditions between guards and forwards, suggesting divergent neural processing efficiency related to player position when performing a cognitive task involving inhibitory control.

We found that the guards had shorter N2 latencies in the Go conditions and longer N2 latencies in the NoGo conditions as compared to the forwards. The N2 components not only reflect conflict monitoring [14,16,17] but also cognitive processes related to stimuli evaluation and response selection [27,41]. To be more specific, the N2 components have been reported have a relationship to the classification of the stimulation [42,43]. Accordingly, the N2 latency reflects the attentional orientation speed in these mental processes [43]. In the present study, we found that, in comparison to forwards, shorter N2 latencies in the Go condition could be observed in the guards, suggesting that they exhibited faster stimuli evaluation and response selection when confronting the target stimuli. Passing is one of the core skills during the offensive stages of games [44]. Appropriate passes facilitate fluent ball movement and allow the ball to move to the other players based on the most optimal offensive opportunity. Passing skills are required in both set plays and transitions. During the set plays, precise timely passes ensure successful tactical enforcement, while during transitions, passes allow quick ball movement down the court, enabling a fast breaker to beat opponents. It should be noted that the execution of passing skills is, at least in part, similar to the mental and response processes to Go and NoGo stimuli. Conditions that passers usually encounter “to pass” (i.e., move the ball to the right man according to tactics) or “not to pass” (i.e., the discovery of the opposites’ counter movements) appear to resemble to the decision-making processes between “respond” or “don’t respond” when engaging in the Go/NoGo task. Considering that guards are better passers and assisters [19] and that they engage in more passing (especially PG) than forwards [45], performance indicating shorter Go-N2 latencies on the part of guards may reflect their excellent capabilities and efficiencies related to being able to pass whenever an open, appropriate offensive opportunity occurs.

However, the guards had longer N2 latencies than the forwards in the NoGo condition. These findings suggest that the guards require prolonged mental processing when recognizing a need for inhibition. Indeed, the results supported the idea that although there was comparable behavioral inhibition between the guards and forwards, the two groups still showed differences in terms of neural processing efficiency. As mentioned previously, we expected the guards to show superior inhibitory controls compared to forwards owing to their cognitive accommodations to additional environmental interferences (e.g., conflicting situations). Surprisingly, since the NoGo-N2 components reflect conflict detection [14,16,17], it appeared that guards tended to show longer duration for classification and selection of responses when violations of expected actions took place [17,41]. This might point out that, instead of adapting to the extra attentional load, whenever a conflicting situation takes place, these additional interferences might force guards to prolong the time required to evaluate and discriminate between stimuli or situations due to the fact that they are more complex, varied situations [19,26], thus leading to their slower N2 latencies in the NoGo conditions as compared to forwards.

With regards to the P3 components, we found that forwards had larger P3 amplitudes than guards across the Go and NoGo conditions and three electrodes. The P3 components reflect the sequential processes of stimuli evaluation, together with N2 components, as well as the later stages of cognitive processing, including premotor preparation and decision-making [29,30,43]. In these cases, given that the amplitude of the P3 components reflects the amount of attentional resource allocation [30,46], the smaller P3 amplitude pointed out that guards mobilize less attentional resources than forwards during the later stages of premotor cognitive processing. As mentioned above, guards are greater passers and assisters who offensively allocate the ball to the best man in order to create the best offensive opportunities. Notably, offensive strategies in basketball also have an important element, which is time. The team’s decision-making will be constrained by the shot clock [47]. Concerning this limitation, teams are often desperate and forced to take a lower quality offensive option that they are previously unwilling to take when the shot clock is going to expire [47]. Accordingly, players who are forced to shoot under the pressure of dwindling time are apt to make rush shot without consideration of whether it is the best choice or not. It is worth noting that players who execute these kinds of risky shots are usually skilled ball handers and shooters [48], or more specifically, guards (especially PG and SG), who can create shooting opportunities even with augmented offensive loadings [47]. That is, when confronted with such circumstances, guards relative to forwards are more likely to make an unreasonable shot rather than to allocate the ball to gain the best offensive opportunity. This might shed light on the tendency of guards to process the perceived stimulus on the basis of rough evaluation and decision-making, allocating less attention during later mental processes. This was supported by Bianco et al. (2017), who found that fencers had greater P3 amplitudes compared to boxers when engaging in the visual Go/NoGo task, since salient incongruencies in exercise strategies differentiate fencing from boxing, with fencers’ training emphasizing accuracy to avoid making errors and boxers being trained to make as many attacks as possible that do not require extreme precision [11].

### 4.3. Limitations

There were some limitations of this study which must be addressed. First, the comparable accuracy rate and scarcity of errors possibly pointed out the simplicity of the task, which could have led to a failure to find between-group behavioral differences [43]. Further work using cognitive tasks with more challenges (e.g., visuospatial attention) is needed to clarify divergent behavioral inhibition [1]. Second, given that the present research was a cross-sectional study, we cannot exclude the superior neurocognitive performance that results from an inherent predisposition toward such performance [22]. Thus, evaluations of the potential contributions of different positions in open-skilled sports to cognitive functioning via longitudinal experiments are needed to clarify this issue [6]. Lastly, the experimental sampling was not completely random due to both the coaches’ and players’ approvals for collecting data being indispensable, and it could be considered as a non-normal sample set. However, in case the coaches and players agreed, all the qualified players in each team were recruited, which is valid enough to indicate the representation of the data collected.

## 5. Conclusions

In conclusion, we found that guards relative to forwards had comparable RTs and ARs, a shorter N2 latency in Go conditions, a longer N2 latency in NoGo conditions, and smaller P3 amplitudes across conditions and electrodes. The current findings supported the premise that guards (1) are more efficient in terms of target evaluation and response selection related to non-target stimuli, (2) exhibit prolonged duration in response selection and evaluation when dealing with target stimulation, and (3) use fewer cognitive resources during premotor preparation and decision-making as compared to forwards. The differences in neural processing characteristics between the two positions could provide important information related to elite basketball training, as well as to some degree provide a potential reference value when recruiting players.

## Figures and Tables

**Figure 1 brainsci-10-00387-f001:**
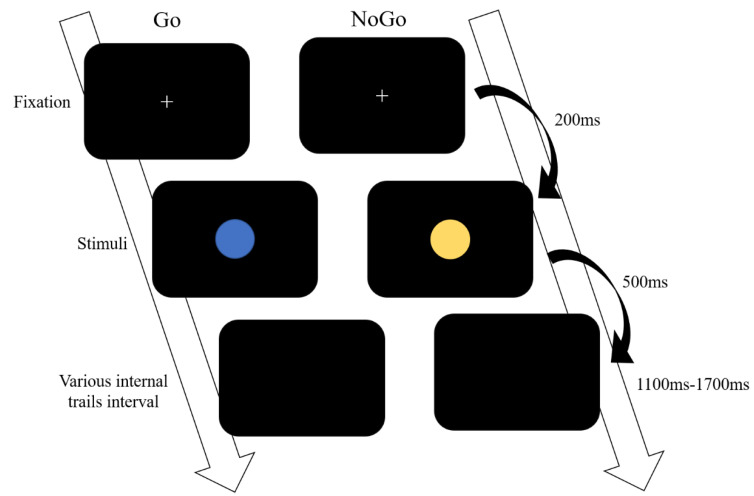
The Go/NoGo paradigm.

**Figure 2 brainsci-10-00387-f002:**
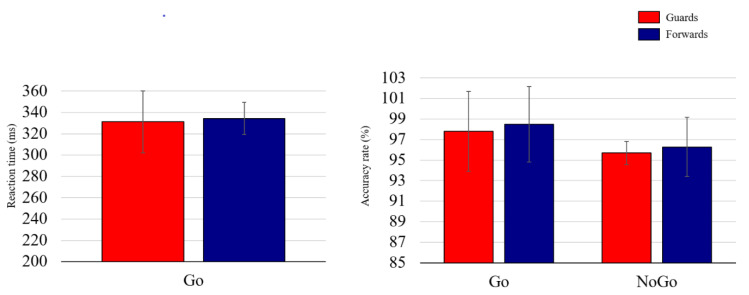
The behavioral performance of the guards and forwards groups.

**Figure 3 brainsci-10-00387-f003:**
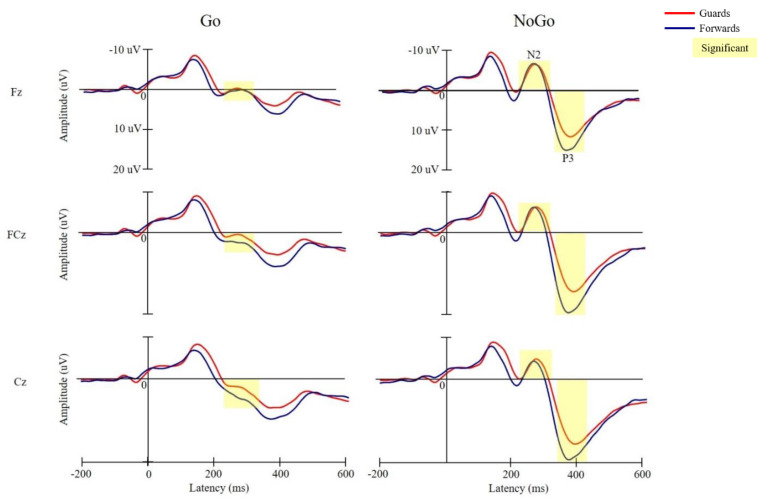
Grand-average ERPs for the Go and NoGo conditions for the Fz, FCz, and Cz electrodes in the guards and forwards groups (red line: guards group; blue line: forwards group).

**Table 1 brainsci-10-00387-t001:** Demographic characteristics (mean ± SD) for the guards and the forward groups.

	Guards (*n* = 27)	Forwards (*n* = 19)	*p*
Age (years)	20.74 ± 1.10	20.26 ± 1.24	0.175
Handiness (R(L))	24 (3)	16 (3)	0.643
Education (years)	14.44 ± 1.37	14.00 ± 0.94	0.199
Height (cm) *	178.59 ± 5.87	190.74 ± 4.65	<0.001
Weight (kg) *	73.54 ± 7.79	86.74 ± 7.33	<0.001
BMI (kg/m^2^)	23.04 ± 1.98	23.80 ± 1.61	0.173
MMSE	28.74 ± 1.13	28.33 ± 1.45	0.318
BDI-II	6.89 ± 6.03	9.00 ± 5.95	0.282
BE (years)	7.22 ± 3.27	6.32 ± 2.65	0.324
TF (times/week)	8.48 ± 2.38	7.53 ± 2.72	0.212
TP (hr/time)	2.60 ± 0.46	2.50 ± 0.50	0.480
VO_2_ max (ml/kg/min)	46.87 ± 3.27	47.79 ± 3.84	0.949

MMSE: Mini Mental State Examination; BDI-II: Beck Depression Inventory, 2nd edition; BE: years of playing basketball experience; TF: Training frequency per week; TP: Training period per session; * *p* < 0.05.

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
