# Peer review of "Behavioral and Cognitive Electrophysiological Differences in the Executive Functions of Taiwanese Basketball Players as a Function of Playing Position"

_brainsci, 2020, doi:10.3390/brainsci10060387_

Round 1

Reviewer 1 Report

This manuscript presents an interesting experiment investigating the behavioral and electrophysiological differences in elite basketball (bball) players. It was expected that different playing positions in bball would recruit a variety of cognitive strategies commensurate to the demands of the position. Similar effects have been seen in other sports like volleyball. The authors used a standard go-nogo task, which measures behavioral intervention, and correlated behavior to the N200 and P300 event-related potentials (ERPs). The N200 is though to index inhibition, mismatch detection, attention. Whereas the P300 is related to more "cognitive" processes like decision making. Contrary to their predictions, the authors found no differences between player positions and performance on the go-nogo task, but they did find differences in the ERPs. The ERP differences are consistent with their hypotheses. This paper is well written, novel, and although the behavioral difference were not found, the manuscript is an important addition to the current science. I agree with the authors that a more sensitive task should be found for bball players. I also think that looking at how the behavioral and electrophysiological measure predict real-world bball metrics would be informative (see my comments at the bottom). Line 48 "based on this..." what is 'this' referring to here? Its not clear. Line 53 - Could you add a line explaining to the uninitiated reader what the N2 ERP means? Something like "the N200 ERP is though to measure mismatch detection in early visual processing, a crucial process for early stage cognitive control." Or cite one study that suggests that the N200 is related to inhibitory control, as that seems important here. You may also want to have one line enplaning what an ERP is. Does the typical reader of sports science know what an event-related potential is? If possible, can you add error bars to the graphs and shade the significant areas on the ERPs? Not necessary, but makes for a more compelling graph and also gives the reader a sense of the effects. Random thoughts - no need to respond to these It does seem that one benefit of sports science is that you have objective, real-world metrics that are collected each game for every player. You should leverage this data and correlate it to your measures. One general comment that strikes me about sports is that it may be true that the average players in one position tend to have, on average, a cognitive strategy commensurate to the task-position. However, this does not necessarily mean that such a strategy leads to elite performance. For example, perhaps a player who uses a novel strategy performs better than the average players. It would be nice to link cognitive strategy and performance to actual performance in the game. Has anybody done thins? Following on my previous half-baked thought, it would be really interesting to look at whether your go-nogo performance and/or N2 ERPs predicts performance on the bball court. Do the "better" players show better cognitive control? I guess my point here is that what you really want to do is develop a metric to distinguish why one player is better than another in bball metrics. So I'd think an analysis that looks at individual performance relative to the whole would make sense (if that is indeed the underlying goal). I have a bit of an ethical knee-jerk to the concluding remark "related to elite basketball recruitment." It seems a little too early to think about using brain activity to decide on whether to recruit a player, especially since this is the first paper to relate bball to ERPs.

Reviewer 2 Report

The study represents an important contribution to the sparse literature on the topic involved, related to cognitive electrophysiological differences in the executive functions of basketball players as a function of playing position. It presents data from an interesting study. The article is well structured and easy to read. However, the Abstract section is not well written, since it does not follow the relevant parts of a study. I believe that the results are sufficiently relevant, although there are a number of issues that would need to be addressed before publication. They are:

- The abstract structure should be improved.

- In terms of writing, the paper relies on some references that (from my viewpoint) could be complemented with some more recent ones. In addition, authors could take into account shorten the introduction section.

- One of the most important requirements would be to explain more or justify the sampling, with all the limitations that it presents. It seems one of the weakest points of the study. At least, it would be interesting if this question were explained in the letter of response to the reviewers. It would be necessary to explain in more detail the recruitment process. Was it randomized? Why not?

- On the other hand, I suggest to the authors to add an "Ethical considerations section".

- Limitations of sampling should be referred.

Overall, it is an interesting study that represents an important contribution to the literature of this field, and it clearly contributes new knowledge to gain insight on it.

Reviewer 3 Report

Thank you for asking me to review the manuscript titled, “Behavioral and cognitive electrophysiological difference in the executive function of basketball players as a function of playing position.” Below are my comments and suggestions.

Title: The title is appropriate. One suggestion is to add “Taiwanese” before basketball players since this helps better describe the sample.

Abstract: The abstract gives an appropriate overview of the manuscript.

Introduction:

  • Line 46 - suggest to change “suffer from” to “experience”
  • Line 53 - thought it is described later on line 163,a brief mention of what N2 amplitudes/latencies measure might help readers early in the paper who are not familiar with ERP
  • Line 91 - suggest “…making successful shots with a ball through their…”
  • Line 100 - change “categories” to “positions”
  • Otherwise, the Introduction does an excellent job of describing the background of the study. I have no further comments here.

Materials and Methods:

  • Line 195 - did the participants practice or exercise prior to the study on the day of the study? I think this is worth mentioning briefly given the effects of exercise on cognition.
  • Similarly, what time of the day did the experiments occur? Brief mention of time of day of the study (morning/afternoon) should be mention given diurnal fluctuations on cortisol that has been shown to affect cognition.
  • Line 197 - reference is needed for the MMSE
  • Line 261 - change to “space bar”
  • Line 266 - BDI-II should be defined in the text and a reference should be added; what is the basketball experience? Is that just the amount of years of basketball played? This should be mentioned briefly.
  • Table 1 - since height and weight were significantly different between the groups, were these variables used as control variables in subsequent analyses?
  • Otherwise, the methods are clear and well-written. Well done.

Discussion

  • Line 366 - “Volleyball” should be one word throughout the manuscript
  • For the Limitations section, it should also be mentioned that the sample was all male and in Taiwanese individuals, so it is unknown if these effects extend to other countries or women.

Overall, the results of the study are very interesting and could provide a neural basis for why guards and forwards have different cognitive abilities. It could also be used as in sports psychology applications to determine if practice lends to increases in cognitive abilities. It will be interesting to see future studies from this research group. The paper is very well-written, and so I have minimal comments. I wish the researchers all of the best with their future endeavors, and hope this review finds them well.
